# The defense island repertoire of the *Escherichia coli* pan-genome

**Dina Hochhauser**, **Adi Millman**, **Rotem Sorek** *

Department of Molecular Genetics, Weizmann Institute of Science, Rehovot, Israel

* rotem.sorek@weizmann.ac.il

## Abstract

It has become clear in recent years that anti-phage defense systems cluster non-randomly within bacterial genomes in so-called "defense islands". Despite serving as a valuable tool for the discovery of novel defense systems, the nature and distribution of defense islands themselves remain poorly understood. In this study, we comprehensively mapped the defense system repertoire of >1,300 strains of *Escherichia coli*, the most widely studied organism for phage-bacteria interactions. We found that defense systems are usually carried on mobile genetic elements including prophages, integrative conjugative elements and transposons, which preferentially integrate at several dozens of dedicated hotspots in the *E. coli* genome. Each mobile genetic element type has a preferred integration position but can carry a diverse variety of defensive cargo. On average, an *E. coli* genome has 4.7 hotspots occupied by defense system-containing mobile elements, with some strains possessing up to eight defensively occupied hotspots. Defense systems frequently co-localize with other systems on the same mobile genetic element, in agreement with the observed defense island phenomenon. Our data show that the overwhelming majority of the *E. coli* pan-immune system is carried on mobile genetic elements, explaining why the immune repertoire varies substantially between different strains of the same species.

## Author summary

Bacteria are commonly infected by viruses called bacteriophages (or phages, for short). To survive phage infection, bacteria employ multiple anti-phage defense systems, many of which were discovered only in recent years. Intriguingly, multiple studies have shown that different strains of the same species can encode completely different sets of defense systems, but the reason for the diversification of defense systems among otherwise nearly identical genomes remained unknown. Here, we systematically characterized defense systems in >1,300 genomes of the model lab strain *Escherichia coli*. We find that anti-phage defense systems are almost always carried on mobile genetic elements such as prophages, transposons and conjugative elements. These elements integrate at specific locations, or "hotspots", within the *E. coli* genome. Different anti-phage defense systems are carried by distinct types of mobile genetic elements that preferentially integrate at specific hotspots, explaining why phage resistance profiles can vary significantly even among closely related

**Data Availability Statement:** The code used to collect and process the data is available at https://github.com/dinahoch/Ecoli-defence-islands.

**Funding:** R.S. was supported, in part, by the European Research Council (grant ERC-AdG GA 101018520), Israel Science Foundation (grant ISF 296/21), the Deutsche Forschungsgemeinschaft (SPP 2330, grant 464312965), the Ernest and Bonnie Beutler Research Program of Excellence in Genomic Medicine, the Minerva Foundation with funding from the Federal German Ministry for Education and Research, and the Knell Family Center for Microbiology. D.H. was supported in part by a fellowship from the Israel Ministry of Absorption. A.M. was supported by a fellowship from the Ariane de Rothschild Women Doctoral Program and, in part, by the Israeli Council for Higher Education via the Weizmann Data Science Research Center.

**Competing interests:** I have read the journal's policy and the authors of this manuscript have the following competing interests: R.S. is a scientific cofounder and advisor of BiomX and Ecophage. The other authors declare that they have no competing interests.

*E. coli* strains. Our findings not only provide a comprehensive view of the distribution of anti-phage defense systems in *E. coli* genomes, but also shed light on the rapid gain and loss of defense systems over short evolutionary timescales.

## Introduction

Bacteria are engaged in a continuous arms race in which they have evolved to defend themselves against the expanding arsenal of weapons at the disposal of phages [1]. To this end, they possess dedicated defense systems that protect against phage infection through a variety of molecular mechanisms [2,3]. Many defense systems used by bacteria were only discovered in the past few years, and it is estimated that many additional anti-phage mechanisms are yet to be discovered [4–8].

Bacterial anti-phage defense systems were shown to be non-randomly distributed in microbial genomes [6,9,10]. Such systems were observed to frequently co-localize in bacterial and archaeal genomes, forming so-called "defense islands": genomic regions in which multiple defense systems cluster together [6,9,10]. The tendency of defense genes to reside next to one another has enabled the discovery of dozens of novel phage resistance systems based on their genomic presence next to known defense systems [4,6,7,11–17].

Although defense islands have served as a remarkably useful tool for the discovery of new defense systems, reasons for the genomic co-localization of defense systems and the nature of defense islands themselves remain poorly understood. Recent evidence suggests that defense systems are frequently carried on mobile genetic elements (MGEs). These include integrative conjugative elements (ICEs) [18], transposons [19], prophages and phage satellites [5,8]. It was shown that these MGEs can possess dedicated hotspots for carrying multiple anti-phage defense systems. Furthermore, several independent studies have demonstrated that MGEs carrying defense systems were directly responsible for differential phage resistance profiles in closely related strains of *Vibrio cholera* and *V. lentus* [18,20]. It has been hypothesized that anti-phage defense systems carried by MGEs participate in inter-MGE warfare and play a role not only in defending the host bacterium against invading phages, but also in protecting resident MGEs against invading MGEs [21,22].

In the current study, we set out to map and investigate the repertoire of mobile defense systems in the *Escherichia coli* pan-genome. *E. coli* is the most well characterized model organism for bacteria-phage interactions, but the arsenal of defense systems in its genome and their preferred mode of mobilization have never been studied thoroughly. By analyzing over 1,300 *E. coli* genomes, we demonstrate that defense systems are almost always carried by MGEs. MGEs carrying defense systems have a marked preference of defensive cargo, as well as preferred integration hotspots within the *E. coli* genome, explaining the considerable variation in phage resistance observed between closely related *E. coli* strains. Our analysis forms a repository of defense islands in *E. coli* strains, a database that may serve as a resource for the discovery of new defense systems in the future.

## Results

In order to find hotspots for integration of defense system-containing mobile elements in the *E. coli* pan-genome, we examined 1,351 *E. coli* genomes downloaded from the Integrated Microbial Genomes (IMG) database [23]. Each genome was scanned for regions containing genes involved in anti-phage defense, searching for mobile regions present in some genomes but missing from others (Methods, Fig 1A). We then mapped these mobile regions to the

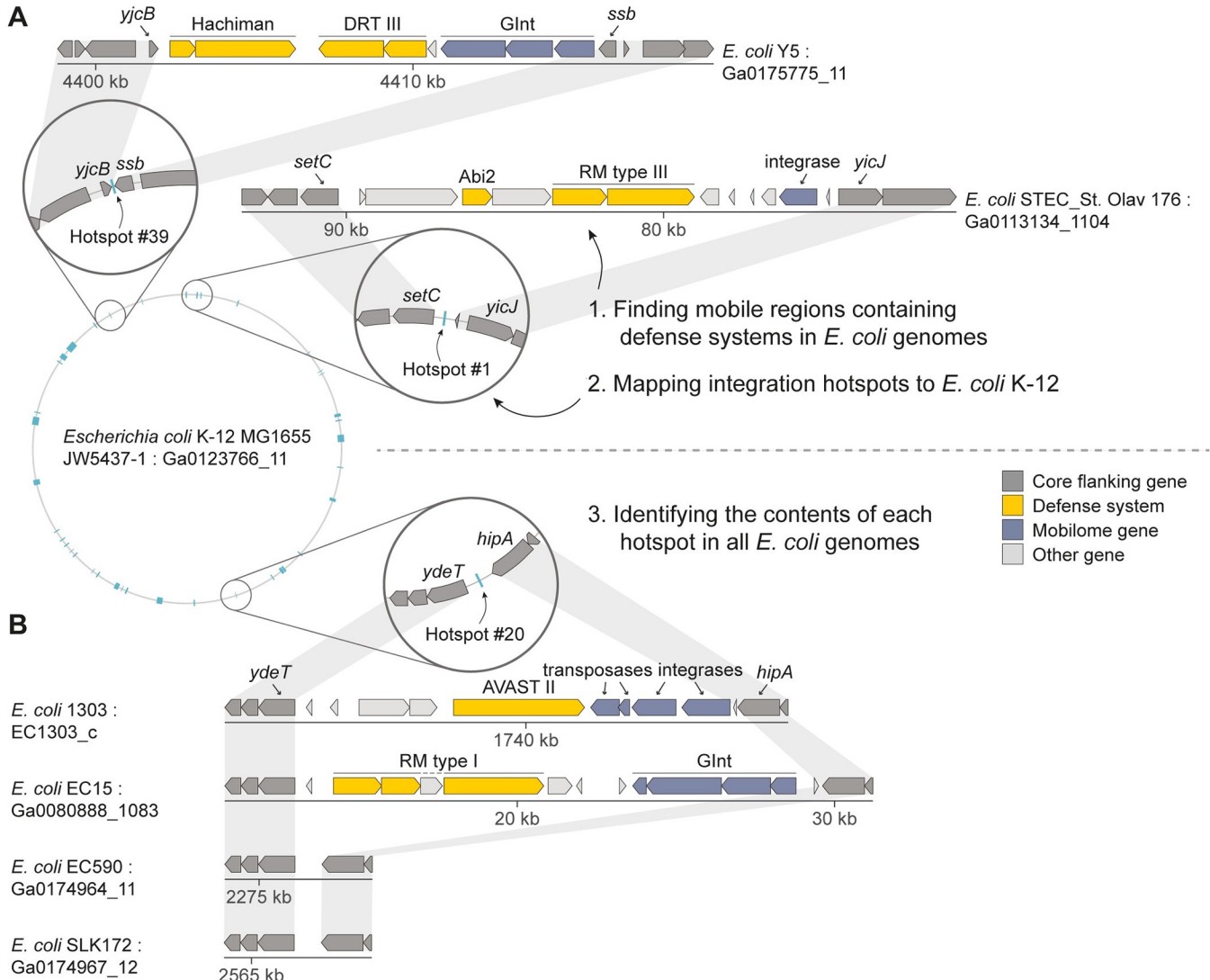

**Fig 1. Schematic of the defense island search approach employed in this study.** (A) Regions containing defense systems in 1,351 *E. coli* genomes were mapped to the *E. coli* K-12 genome based on flanking core genes, identifying hotspots for integration of defense-carrying mobile elements. (B) Each hotspot was then searched for in all other *E. coli* genomes in order to characterize the hotspot occupancy in the *E. coli* pan-genome. The accession number of each genomic scaffold in the IMG database [23] is shown. Gray shading indicates conservation of core genes flanking the integration position of mobile islands. Known defense system genes are marked in yellow. GInt, Genomic Island with three Integrases.

reference genome of *E. coli* strain K-12 MG1655, a commonly used laboratory strain whose genome is well characterized. These defense system-containing mobile islands mapped to 41 discrete hotspots, most of them empty (unoccupied) in the reference *E. coli* K-12 genome (S1 Fig and S1 Table).

To understand the occupancy of the 41 hotspots in the *E. coli* pan-genome, we used the core genes immediately flanking each hotspot in the *E. coli* K-12 reference genome to map these integration hotspots in the 1,351 downloaded genomes (Fig 1B). With a few exceptions, a given integration hotspot was unoccupied in the majority of genomes in which it was detected, with a median of 8% occupancy per hotspot (Fig 2A and S1 Table). An exception was the type I-E CRISPR-Cas locus at what we defined as hotspot #7 (S1 Fig), which appears to be part of the core genome of *E. coli* and is not found on a mobile genetic element [24]. This locus

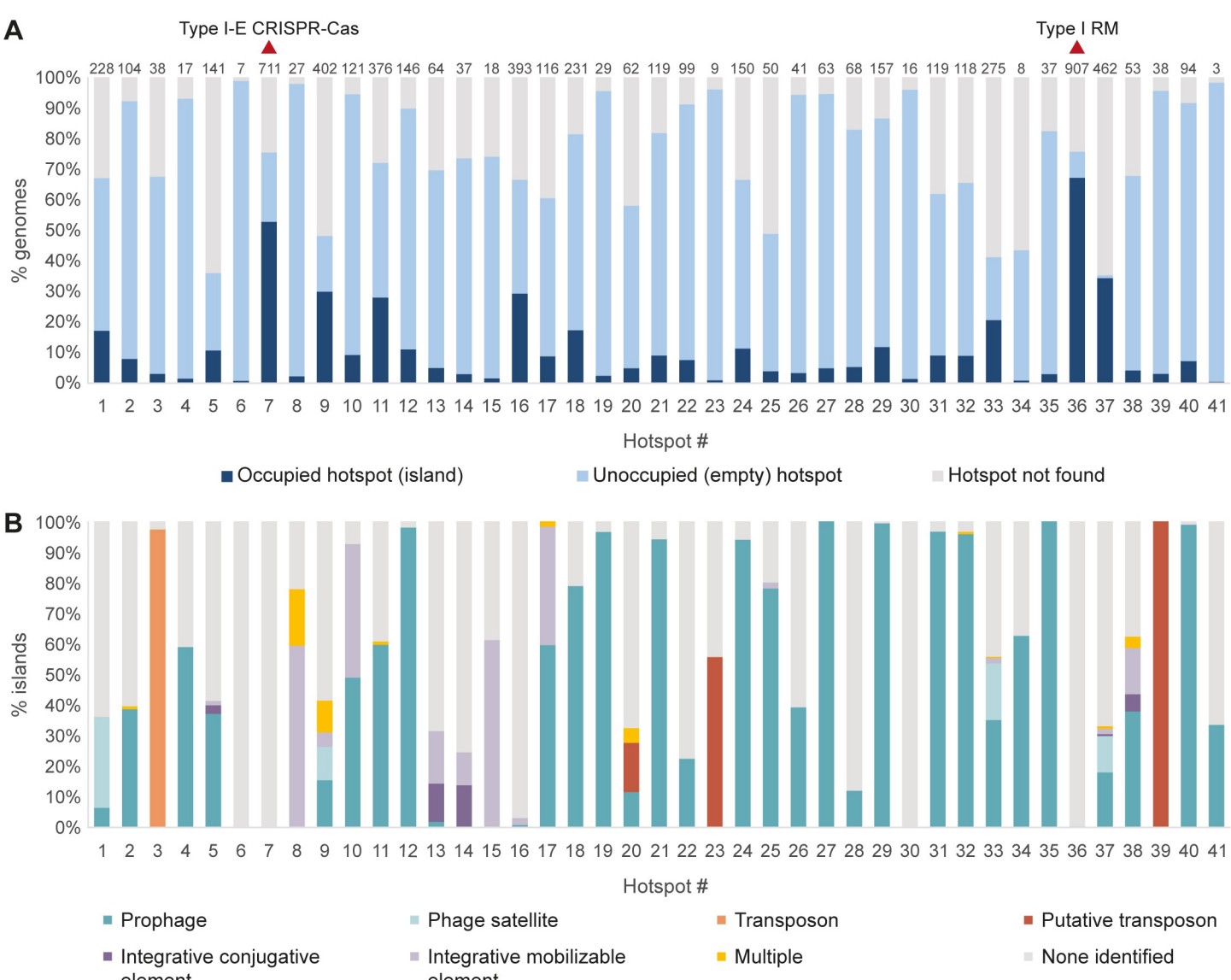

**Fig 2. Occupancy of defense hotspots.** (A) Bar graph showing the occupancy of each integration hotspot among 1,351 *E. coli* genomes analyzed. The number above each bar indicates the number of genomes in which the hotspot was found occupied. "Hotspot not found" (gray) indicates that one or both core flanking genes were not found in the relevant genome. (B) Nature of mobile genetic elements (MGEs) integrated at hotspots identified in this study. Multiple, analysis of genes in the integrated element suggests a combination of multiple types of MGE.

was present in ~70% of the genomes that we analyzed, while in the remaining ~30% it was degraded, explaining why it was identified as a variably occupied hotspot in our initial analysis. Another locus that was often occupied was the type I restriction-modification (RM) locus at hotspot #36, which was flanked by a transposable element and occasionally included additional defense systems such as Druantia and type IV RM systems (Fig 2A and S2 Table).

As expected from a recent analysis of the *E. coli* pan-genome [25], the defense system-containing mobile islands that we found mostly consisted of well characterized MGEs including prophages, phage satellites, transposons, integrative conjugative elements (ICEs), and integrative mobilizable elements (IMEs) (Fig 2B and S2 Table). Prophages were the most abundant MGE type carrying defense systems (Fig 2B).

Many of the hotspots that we identified were previously described as integration positions for known MGEs. Specifically, 18 of the hotspots were within tRNA loci in the *E. coli* genome, which are commonly used by prophages and other MGEs as integration hotspots [26]. Some hotspots were occupied by only a single type of MGE. For example, hotspot #29 was occupied only by phages of the Felsduovirus taxonomy that integrated within the small RNA rybB [27]. We found 109 *E. coli* genomes in which this hotspot was occupied by similar prophages of the Felsduovirus genus, each of which carried up to two defense systems (S2 Fig and S2 Table). As another example, the Tn7-like transposon Tn*6230* integrated only at hotspot #3 between the genes *yhiN* and *yhiM*, as previously documented for this family of transposons [28,29]. On the other hand, some hotspots in our dataset were occupied by a diverse variety of MGEs in different genomes. For example, hotspot #9, which occurs within the tmRNA locus, could contain integrated prophages of multiple taxonomical groups, P4-like phage satellites, integrative mobilizable elements, and transposons (S2 Table). This is explained by the widespread use of the tmRNA gene as an integration position for different MGEs, with multiple integrase subfamilies having independently evolved to integrate at this position [26,30].

Many MGEs that carry defense systems cannot independently mobilize between genomes but are rather "parasites" of autonomously mobilizable MGEs. For example, phage satellites are known to carry anti-phage defense systems that they package into capsids of "helper" phages along with the helper phage DNA [5,31,32]. P4-like phage satellites commonly integrated within tRNA genes at hotspots #1, #9, #33 and #37 (Fig 2B and S2 Table). Integrative mobilizable elements (IMEs) are another form of "parasitic" MGE in which defense systems were commonly found (Fig 2B and S2 Table); these MGEs do not constitute full conjugative elements, but hitchhike on other conjugative elements for transfer between species [33]. We found such IMEs carrying defense systems integrated at multiple hotspots (Fig 2B and S2 Table).

Some mobile elements that carry defense systems did not fall into a specific category of commonly known MGEs. Some of these mobile elements were characterized by genes annotated as "phage integrase", or multiple genes annotated as integrases or recombinases, but no additional phage genes were detected in the island (Fig 3A–3C). The presence of these islands in only a subset of genomes suggests that they are somehow mobile, but it is not clear how such elements can mobilize between genomes in the absence of additional known mobility genes (Fig 3A–3C). It is possible that these islands constitute yet unidentified transposons or other MGEs. Indeed, the recently described Tn*6571*-family transposon termed GInt (<u>G</u>enomic <u>I</u>sland with three <u>Int</u>egrases), which comprises three putative integrase genes and a small helix-loop-helix protein [34–36], was identified in our analyses as a defense-carrying element integrated at multiple hotspots (Fig 3D and S2 Table). Alternatively, integrase-only defense-carrying islands may represent yet uncharacterized types of "satellite" elements that parasitize other MGEs for their mobilization.

Overall, we detected 87 types of known defense systems in the hotspots identified in this study (Fig 4). We found that the same type of MGE can carry different sets of defense systems when integrated in different genomes (Figs 3, S2 and S3 and S2 Table). Felsduovirus prophages integrated at hotspot #29, for example, contained a large diversity of defense systems at dedicated positions in the phage genome (S2 Fig). Similarly, a dedicated position within an ICE element that preferentially integrates at hotspots #13 and #14 could contain CBASS, Gabija, Hachiman, Lamassu, retron, and additional systems (S3 Fig and S2 Table). Indeed, it was previously demonstrated that phages and other MGEs carry defense systems at dedicated locations in their genomes [5,8,18].

Some types of defense systems showed preference to be carried by a specific type of mobile genetic element, or to be integrated at specific hotspots (Fig 4). For instance, the *bstA* gene, which encodes an abortive infection protein, was found only in prophages integrated at

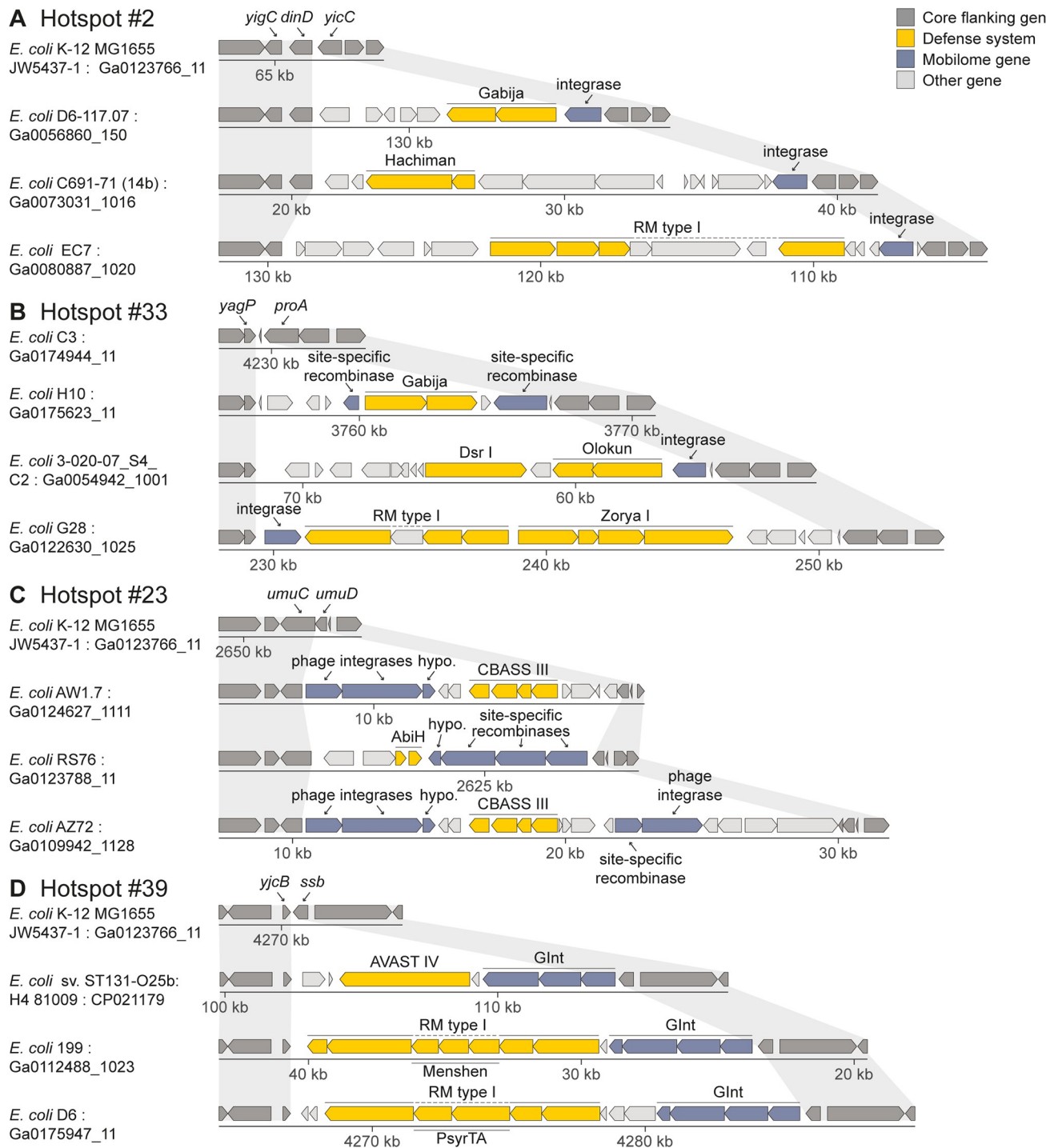

**Fig 3. Examples of mobile islands carrying defense systems with unclear mechanisms of mobilization.** These islands typically contain integrase or recombinase genes but lack other known mobility genes. (A) Selected examples of integrase-only mobile elements integrated at hotspot #2. (B) Selected examples of integrase-only mobile elements integrated at hotspot #33. This hotspot is occupied in *E. coli* K-12. (C) Selected examples of hotspot #23, comprising defense systems associated with multiple integrases. (D) Selected examples of hotspot #39 occupied by GInt, a newly described Tn*6571*-family transposon [34–36]. Gray shading indicates conservation of core genes flanking the integration position. RM, restriction-modification; hypo., hypothetical gene; Gabija, Hachiman and Zorya are defense systems described in [6]; AVAST was described in [4]; CBASS was described in [13]; Olokun, Menshen, and PsyrTA were described in [7]. Gene symbols of flanking core genes are indicated for each hotspot.

hotspot #32 (Fig 4). This gene is naturally silenced by a cognate anti-BstA (*aba*) DNA element, providing defense against multiple phages that lack the *aba* element [37]. Ten instances of BstA-carrying lambda-like prophages were identified within hotspots, all of which integrated within the tRNA$^{Arg}$ gene at hotspot #32. Similarly, the BREX system [11], which appears in 49 islands in our set, was present only at hotspots #1 and #37; the abortive infection system AbiEii [38] was found only at hotspots #5, #33, and #37; and mobile elements carrying the Wadjet and Zorya defense systems showed preference for integration at hotspot #37 (Figs 4 and 5).

Hotspot #37 was found to contain an exceptionally high diversity of defense systems (Figs 4–6). This hotspot was occupied in nearly all (97.3%) *E. coli* genomes, and when occupied, it typically (97.4%) contained at least one defense system, with a total of 31 defense system types identified across different genomes (Figs 4–6). This suggests that hotspot #37 represents a genomic position dedicated to defense systems in *E. coli*. However, the mode of mobilization of these systems between genomes could not be readily determined. While in some cases hotspot #37 contained prophages, P4-type phage satellites or IMEs, the majority (68.0%) of cases did not have any detectable MGEs, although many contained integrase or recombinase genes (Fig 6 and S2 Table). Notably, a recent study showed that *Pseudomonas aeruginosa* genomes encode two highly diverse hotspots that seem to be similarly dedicated to carrying defense systems, with some cases showing no identifiable modes of mobilization [39].

To understand the contribution of MGEs to the defense repertoire of *E. coli*, we next examined 190 *E. coli* genomes defined as "finished" in the IMG database [23], i.e., their genomes are completely assembled with no gaps. A given finished genome had, on average, 10.2 of the 41 hotspots occupied with an integrated element, but only a subset of these (between one and eight, 4.7 on average) contained known defense systems (S3 Table). Analyzing the defense system content of the main chromosome of each of these genomes using DefenseFinder [40] revealed a total of 1,577 defense systems. Of these, 1,429 (90.6%) were found at the 41 hotspots mapped in the current study (S3 Table). Defense systems frequently (58.9% of cases) co-localized with at least one other system on the same island (S3 Table), conforming with the previously observed tendency of defense systems to genomically co-localize [6,9,10], but also showing that defense systems frequently appear alone [8,40]. Together, these data suggest that the overwhelming majority of the chromosomal defense system repertoire of the *E. coli* pan-genome is carried on mobile genetic elements that preferentially integrate at a discrete set of defined genomic positions.

## Discussion

For decades, *E. coli* has been the workhorse for studies on interactions between bacteria and phages [41]. Many bacterial defense strategies have been discovered, and the mechanisms by which phages evade these defenses extensively studied, using *E. coli* as a model organism [41–43]. The dataset of mobile elements carrying defense systems that we have collected provides a reference point for defense system occupancy in the pan-genome of *E. coli* and may serve as a resource for future studies aimed at examining the *E. coli*-phage relationship.

Our data show that the vast majority of chromosomally-encoded defense systems in the *E. coli* pan-genome are carried on mobile genetic elements, explaining the immense variability in presence and absence of these defense systems among different strains of the same species [1]. These data are consistent with recent studies showing that in strains of *Vibrio* species, MGEs carrying defense systems comprise the majority of accessory genes in the pan-genome [18,20]. Indeed, recent studies have detected new defense systems specifically in *E. coli* prophages [8,37] or have relied on defense-rich hotspots within prophage genomes to reveal new defense systems [5]. The presence of defense systems within MGEs has been described as a "guns for

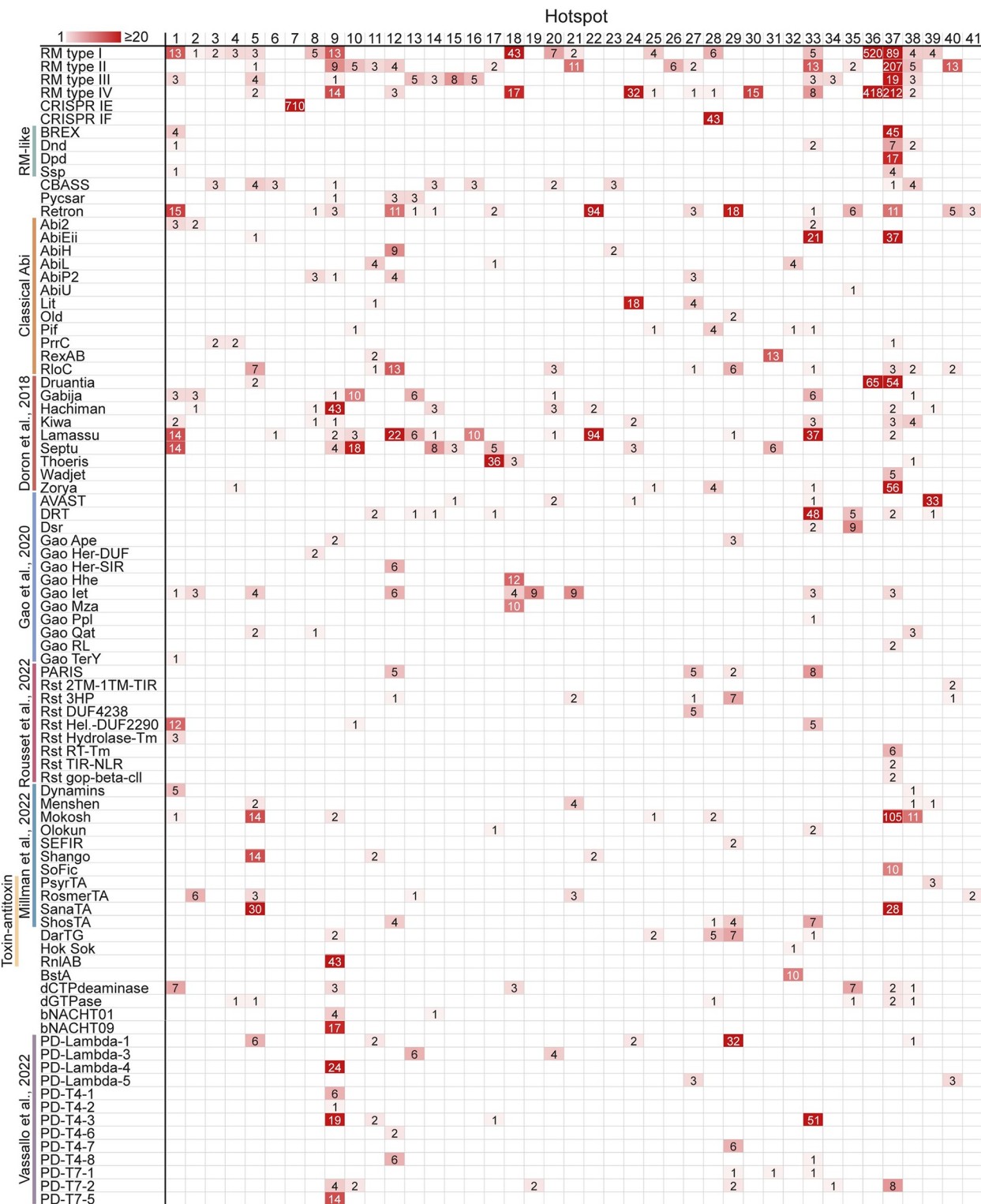

**Fig 4. Defense system occupancy at 41 hotspots in the *E. coli* pan-genome.** Numbers indicate the occurrences of each defense system within each hotspot, with red shading corresponding to the frequency.

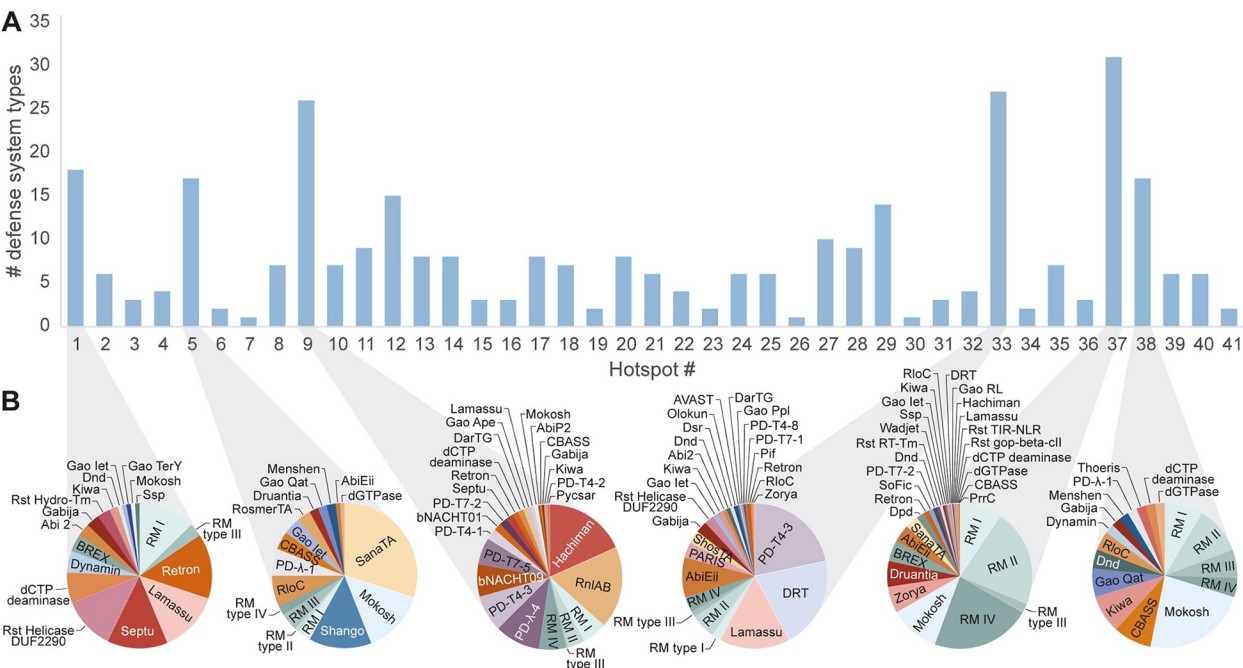

**Fig 5. Diversity of defense systems at *E. coli* hotspots.** (A) Number of different defense systems found at each of the 41 hotspots mapped in this study. (B) Defense system distribution for a selected set of the most diverse hotspots.

hire" scenario, in which defense systems may promote the persistence of MGEs in the host genome by providing a fitness advantage against other invading MGEs [21].

The hotspots that we identified demonstrated a wide variety of occupancy rates, with some hotspots very poorly occupied. Specifically, hotspots #6, #23 and #41 were each occupied in less than 1% of genomes. These hotspots could represent integration positions for rare site-specific MGEs or neutral landing grounds for MGE integration [44]. In contrast, hotspot #37 was highly occupied, with diverse defense systems present in almost all genomes in which it was detected. This position seems to represent a dedicated defense hotspot that may play a role in promoting defense system diversity within the *E. coli* pan-genome, as was also observed in strains of *P. aeruginosa* [39].

Many coliphages are strain-specific, infecting only a subset of *E. coli* strains [45]. With our map of defense systems localized to *E. coli* defense islands, it will be possible to ask whether mobilization of defense-containing elements leads to resistance against particular phages and to potentially identify specific defense systems preventing phage infection. Moreover, with our accurate mapping of the boundaries of defense-containing mobile islands, it will be possible to search these islands for yet-undiscovered defense systems, thus empowering future studies aimed at expanding the current knowledge of bacterial defense.

Our study, as well as recent studies by others [5,8,18], demonstrates that defense systems cluster within specific regions in MGEs, yet the reasons for this clustering are not entirely clear. It is likely that this aggregation of defense systems within mobile elements provides fitness benefits to recipient bacteria living in a phage-rich environment [1]. It has also been suggested that synergism between defense systems may promote their co-localization within and co-transfer between genomes [22,46,47]. Genes involved in bacterial pathogenicity are frequently mobilized between bacteria on "pathogenicity islands", which constitute MGEs carrying clusters of pathogenicity genes [48]. Similarly, multiple antibiotic resistance genes tend to

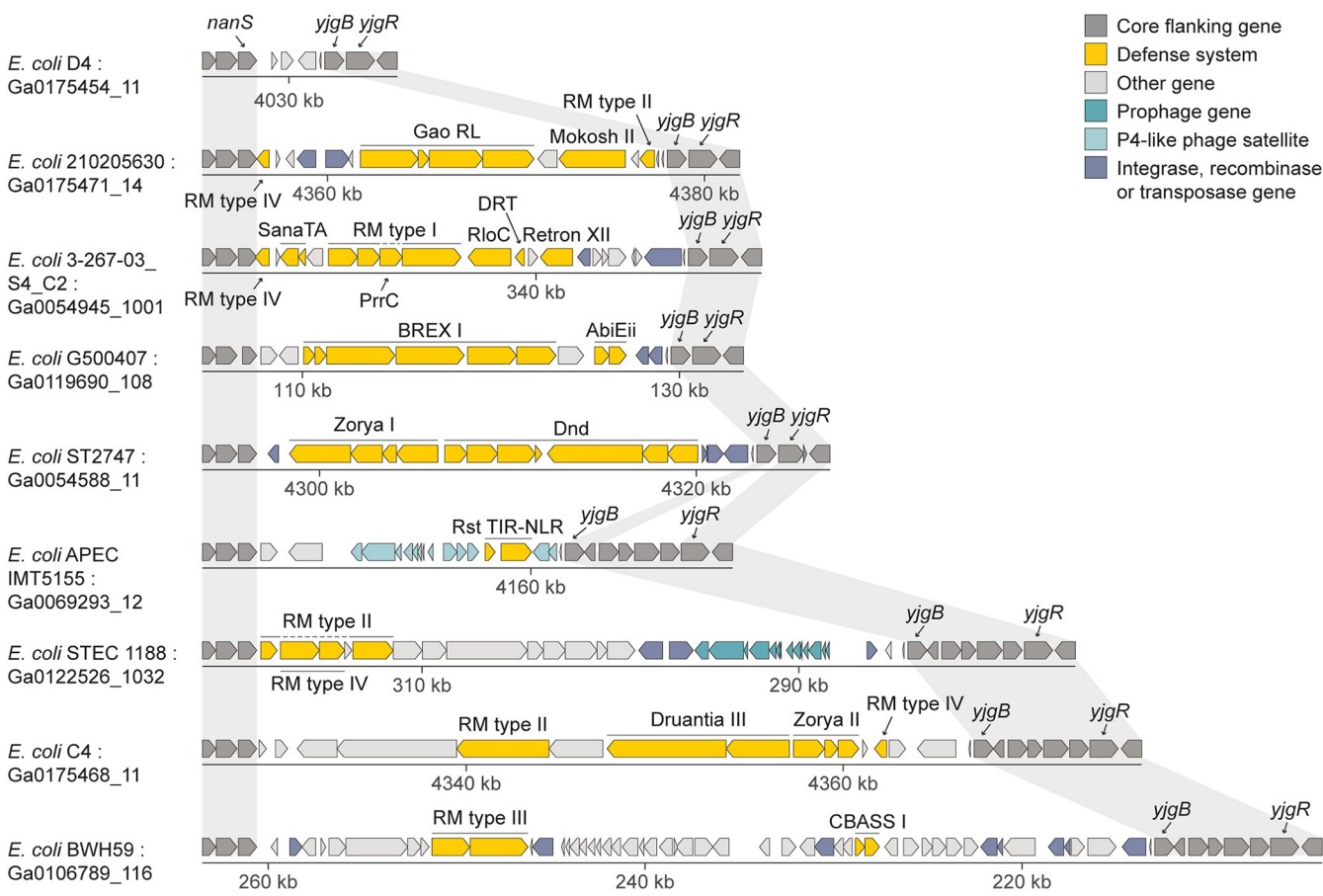

**Fig 6. Hotspot #37 contains an extraordinary diversity of defense systems.** This hotspot is occupied in *E. coli* K-12.

cluster on plasmids and other MGEs [49,50]. It is possible that the same evolutionary forces acting to aggregate pathogenicity and antibiotic resistance genes on the same mobile element could act to promote defense system aggregation within defense islands. Understanding the exact nature of these evolutionary forces, as well as the benefits and costs of hosting mobile elements encoding defense islands, awaits future studies.

## Methods

### Defining mobile islands that carry defense systems in the *E. coli* genome

Prokaryotic genomes were downloaded from the Integrated Microbial Genomes (IMG) database [23] in October 2017, and all proteins from these genomes were grouped by sequence similarity using MMSeqs2 [51] to form clusters of homologs, as previously described [7]. The subset of *E. coli* genomes in the downloaded dataset was then further analyzed. Genome assemblies that were highly fragmented and comprised of more than 200 contigs were discarded. The following defense systems were identified in the *E. coli* genome dataset based on cluster annotation, as previously described [7]: RM, CRISPR, BREX [11], CBASS [13], DISARM [12], Dnd [52], Druantia, Gabija, Hachiman, Kiwa, Lamassu, Septu, Shedu, Thoeris, Wadjet, Zorya [6], pAgo [53], retrons [14], STK2 [54], Aditi, Azaca, Bunzi, Dazbog, Dodola, Menshen, Olokun, Shango [7], dCTP deaminase [16], DSR1, DSR2, Gao Qat, Gao SIR2-HerA,

RADAR [4], PrrC, and Abi proteins (pfams PF07751, PF08843, PF09848, PF10592, PF14253, PF14355).

To determine the boundaries of each putative mobile island that carries defense systems, the genomic regions upstream and downstream of each system were scanned until reaching "flanking genes" that are part of the core genome, i.e., belonging to protein clusters found in more than 80% of *E. coli* genomes. Defense system-carrying mobile islands were defined as defense system-containing regions of at least ten genes which were present in some genomes but absent in others, i.e., their flanking genes were found adjacent to one another in at least one *E. coli* genome.

## Mapping defense system-carrying mobile islands to the *E. coli* K-12 reference genome

In order to precisely map integration hotspots of mobile islands that carry defense systems in the *E. coli* K-12 reference genome, clusters of the flanking genes of each island were compared to clusters in the genome of *E. coli* K-12 MG1655 JW5437-1 (IMG genome ID 2687453259) until a syntenic region was found between the genomes. This was defined as a block of five flanking consecutive genes in the same order and with the same respective clusters as five consecutive genes in the K-12 genome. For cases of gene deletions or duplications in the flanking regions of the islands, these hotspots were manually inspected to define the exact integration hotspot and the precise flanking genes (S1 Table).

## Using K-12 flanking genes to determine mobile island occupancy in all *E. coli* genomes

Each hotspot was searched for in all analyzed *E. coli* genomes using the two genes immediately flanking the hotspot in the K-12 genome (S1 Table). To exclude fragmented contigs, only contigs with more than 20 genes were considered. For cases where the immediately flanking genes were not found in the target genome, or in cases where multiple instances of immediately flanking genes were found, a window of ten genes on either side of the flanking genes in K-12 was searched for in the target genome, requiring at least five of the genes matching in gene order and cluster identity. If multiple matches were found, the closest set of upstream and downstream flanking genes were selected to define the hotspot. Both flanking regions were required in the same contig to declare a hotspot. The resulting islands at these hotspots were filtered for those containing 200 or fewer genes and an individual hotspot was defined as "empty" if it comprised three or fewer genes.

## Clustering mobile islands by sequence similarity and manually curating representative islands

In order to remove redundancy in the resulting islands, the nucleotide sequences of the islands at each hotspot were clustered using the cluster module in MMSeqs2 release 12-113e3 [51], with the parameters --cov-mode 0 -c 0.8 --cluster-mode 2 --min-seq-id 0.6 -s 8 --threads 1. The MMSeqs2-determined representative sequence from each cluster was taken as the representative sequence of the island. All representative island sequences were manually curated to adjust their flanking genes where necessary (e.g., in the cases of pseudogenized or repeating genes).

### Identifying mobile genetic elements in islands

Islands were inspected for genes associated with mobile genetic elements (MGEs). MGE type was determined as described below. In cases where two MGE types were clearly integrated within the same island, the island was recorded as comprising multiple MGE types.

**Prophages and their satellites.**   Prophages were identified using PHASTER [55] analysis of the nucleotide sequences of each island. A phage hit was only considered if the island had more than one gene that matched the phage. When PHASTER identified intact prophages, the taxonomy of the phage hit was recorded using NCBI classification (S2 Table).

Phage satellites were detected using SatelliteFinder (Galaxy Version 0.9) [56] analysis of the amino acid sequences of genes in each island. P4-like satellites were only considered if they were predicted to be of types A, B or C, and PICI satellites if predicted to be of types A or B, per the definitions in ref. [56]. Manual inspection of islands annotated to contain PICI satellites revealed several of these to be intact Uetakevirus prophages; the annotation was changed accordingly.

When PHASTER and SatelliteFinder gave overlapping predictions, the SatelliteFinder prediction was used and the prediction was checked by manual inspection.

**Integrative conjugative elements and integrative mobilizable elements.**   The amino acid sequences of genes in each island were submitted to CONJscan with default parameters (Galaxy Version 2.0+galaxy1) [57]. This identified both integrative conjugative elements (ICEs) and also integrative mobilizable elements (IMEs) that can hijack ICEs. In addition, islands were searched for genes annotated as VirB5 and VirB6, which are known components of non-canonical IMEs that lack a relaxase gene [33].

**Transposons, Genomic Islands with three Integrases (GInts) and putative transposons.**   Tn7-like transposon genes were classified by searching for genes annotated as "TniQ" and manually verifying the presence of the transposon. Homologs of Genomic Islands with three Integrases (GInts) genes from ref. [34] were searched for using BLAST protein search with default parameters. Multi-integrase cassettes were classified as GInts if they had at least one integrase gene with homology to a GInt integrase (e-value of less than 1e-05) and a similar genomic organization of three integrase genes and a short hypothetical gene. Multi-integrase systems not annotated as transposons or GInts were classified as putative transposons.

### Identifying defense systems in mobile islands

DefenseFinder [40] and PADLOC [58] were utilized to identify known defense systems in each mobile island. Amino acid sequences of genes in all 1,351 *E. coli* genomes were submitted to DefenseFinder release version 1.2.0 [40]. Genes predicted to be part of multiple different defense systems were inspected manually for proper annotation. Amino acid sequences and gff3 files of genes in each island were submitted to the PADLOC web server v1.1.0 [58] with defense systems included in padlocdb v1.4.0. Systems annotated by PADLOC as "[system]_other" were excluded, since these represent partial or separated defense systems. When two overlapping systems of the same type were predicted by the two tools, all constituent genes were considered part of this system.

### Mapping defense systems found in finished genomes to the *E. coli* K-12 reference genome

Amino acid sequences of genes in the main chromosomes of all finished *E. coli* genomes were similarly analyzed using DefenseFinder release version 1.2.0 [40]. Defense systems were

mapped to the *E. coli* K-12 reference genome as described above. These were then manually examined to identify exactly where in the genome they were integrated. If the integration position constituted a hotspot but this had not been detected due to deletion of flanking core genes on one or both sides, this was manually recorded in S3 Table.

## Supporting information

**S1 Fig. Genomic map of integration hotspots in the *E. coli* K-12 reference genome.** Numbers outside the ring indicate hotspot number. Flanking core genes for each hotspot are indicated on the inside. Blue ticks indicate the position of the hotspot in the *E. coli* K-12 reference genome, with thicker ticks reflecting hotspots that are occupied in K-12. When a given hotspot is occupied in the K-12 genome, tick thickness is proportional to the size of the island inserted at the hotspot.
(TIF)

**S2 Fig. Phages of the Felsduovirus genus integrated at hotspot #29 carry a diversity of defense systems.** Shown are multiple examples of Felsduoviruses that integrate at hotspot #29. Defense systems are marked in yellow. Genome similarity was visualized using Clinker [1]. 1. Gilchrist CLM, Chooi YH. clinker & clustermap.js: automatic generation of gene cluster comparison figures. Bioinformatics. 2021;37: 2473–2475.
(TIF)

**S3 Fig. An integrative conjugative element carrying diverse defense systems integrates at dedicated hotspots.** Instances of this element carry defense systems and are integrated at hotspots #13 and #14. Grey shading indicates conservation of core genes flanking the integration position at hotspot #14, purple shading indicates conservation of the integrative conjugative element (ICE) at these loci. Only part of the island is shown for space constraints.
(TIF)

**S1 Table. Hotspots identified in the reference *E. coli* MG1655 K-12 genome.**
(XLSX)

**S2 Table. Representative islands integrated at the hotspots identified.**
(XLSX)

**S3 Table. Defense systems detected in finished *E. coli* genomes.**
(XLSX)

## Acknowledgments

We thank the Sorek laboratory members for comments on earlier versions of this manuscript.

## Author Contributions

**Conceptualization:** Dina Hochhauser, Rotem Sorek.

**Data curation:** Dina Hochhauser, Adi Millman.

**Formal analysis:** Dina Hochhauser.

**Funding acquisition:** Rotem Sorek.

**Investigation:** Dina Hochhauser.

**Methodology:** Dina Hochhauser.

**Project administration:** Rotem Sorek.

**Software:** Dina Hochhauser.

**Supervision:** Rotem Sorek.

**Validation:** Dina Hochhauser.

**Visualization:** Dina Hochhauser.

**Writing – original draft:** Dina Hochhauser, Rotem Sorek.

**Writing – review & editing:** Dina Hochhauser, Rotem Sorek.

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
