## [Decision Letter · Decision Letter 0]

6 Mar 2023

Dear Dr Sorek,

We are pleased to inform you that your manuscript entitled "The defense island repertoire of the Escherichia coli pan-genome" has been editorially accepted for publication in PLOS Genetics. Congratulations!

Yours sincerely,

Benjamin Schwessinger

Academic Editor

PLOS Genetics

Eva Stukenbrock

Section Editor

PLOS Genetics

Comments from the reviewers (if applicable):

Thank you very much for submitting this very comprehensive manuscript to PLOS Genetics. We are happy to inform you that your manuscript is accepted as is. While one reviewer suggested additional analysis, the editor does not think these additional analysis are needed for publication in PLOS Genetics. The comments are included to inspire future research.

Reviewer's Responses to Questions

**Comments to the Authors:**

Reviewer #1: This study by Hochhauser et al. aimed to better understand the nature and distribution of bacterial defense systems against phages, which are nonrandomly clustered within bacterial genomes in "defense islands". The researchers comprehensively mapped the defense system repertoire of over 1,300 strains of Escherichia coli, a widely studied organism for phage-bacteria interactions and generally an important model organism in microbial genetics. They found that defense systems are often carried on mobile genetic elements (MGEs) such as prophages, integrative conjugative elements, and transposons. Furthermore, they identified several dedicated hotspots for the integration of those MGEs in the E. coli genome. Each mobile genetic element type had a preferred integration position but can carry a diverse variety of defensive cargo. Examples of specific hotspots including diverse repertoire of defense systems are presented in the manuscript. The analysis supplies a rich dataset of E. coli defense mechanisms against phages (and possibly other types of invasive foreign DNA). I have no doubt that other scientists will make use of this carefully curated information. The manuscript is very well written, the figures are easy to comprehend and the methods are clearly described. I have no critical comments on this submission.

Reviewer #2: This study from Hochhauser, Millman, and Sorek provides a thorough dissection of the defense hot spots in E coli, providing really valuable information about what kinds of elements move which kinds of systems. What emerges is surely a complex picture of MGEs carrying different defense systems with particular integration sites and biases that are noted. This work is very important, comprehensive and clearly explains the majority of known systems in this organism. The figures are clear and the analysis answered my next question at every turn. I have no suggested changes and think it should be published.

Reviewer #3: In this informative study, the authors comprehensively identify location and content of all “defense islands” in >1,300 strains of E. coli. Somewhat predictably they find that most defense systems are MGE-encoded and that some “hotspots” are associated with specific elements (presumably because their recombinase/integrase is specific for the integration site. The work is thorough and solid, but somewhat uninspiring. I think that several additional interesting biological questions could be asked with these data, and addressing at least some of them will make the paper more interesting.

I have included a list of such questions below:

1. Is there an association between defense system type and MGE type? There are straightforward ways to identify ICEs, Prophages and satellites, and classify them as either one or the other and such data are found to some extent in Table S2. This has potential “added value”, beyond further classification, since some systems are really exclusively anti-phage, while others can defend against ICEs and transposons as well. This may show whether an island-encoded defense is truly a “gun for hire” or in some cases just used against direct competitors, and we know what are some direct competitors because they tend to compete for the same hotspot site.

2. Is there something special about non-MGE defense islands (in the strict sense, with no integrase or recombinase/transposase in them), so they tend to encode other systems integrated at more conserved loci? Not integrated at tRNA genes?

3. Looking at figure it is obvious that nearly always one or both flanking regions are absent, implying that the regions in the genome where defense islands are found tend to be variable even before something gets integrated there or that these regions are unstable (for example due to recombination between IS elements). One can distinguish between the two scenarios by looking at the location of flanking genes in the complete genomes with respect to the origin and terminus, which is anyway interesting in this context.

4. One advantage for MGEs in integrating into tRNA genes is that these are relatively conserved in sequence and often found in all strains. However, most islands here have non-tRNA integration sites, including some that are very often occupied. Looking at the non-tRNA integration sites 50-100 bases upstream and downstream are there interesting or common sequence features? How conserved across strains are non-tRNA sites in terms of sequence identity?

5. Is there a higher defense island density in E. coli strains from non-human sources (animal, water, etc), or in non-pathogens vs. pathogens?

6. For CRISPR-Cas systems an inverse association was observed between the number of prophages and the presence of systems, etc. Is there such an inverse association between prophage/satellite/ICE number and defense system number in E. coli?

A minor comment is that reference 60 deals with colibactin biosynthesis island not the “high pathogenicity island” common to many E. coli strains that is ICE-derived. Colibactin was shown to induce DSBs and later also prophage excision and entry into lytic cycle. Author should revise the text accordingly.

**Have all data underlying the figures and results presented in the manuscript been provided?**

Reviewer #1: Yes

Reviewer #2: Yes

Reviewer #3: Yes

PLOS authors have the option to publish the peer review history of their article (what does this mean?). If published, this will include your full peer review and any attached files.

Reviewer #1: No

Reviewer #2: **Yes: **Joseph Bondy-Denomy

Reviewer #3: No

**Data Deposition**

http://datadryad.org/submit?journalID=pgenetics&manu=PGENETICS-D-23-00150

**Press Queries**

---

## [Editor Report · Acceptance letter]

3 Apr 2023

PGENETICS-D-23-00150 

The defense island repertoire of the *Escherichia coli* pan-genome 

Dear Dr Sorek, 

We are pleased to inform you that your manuscript entitled "The defense island repertoire of the *Escherichia coli* pan-genome" has been formally accepted for publication in PLOS Genetics! Your manuscript is now with our production department and you will be notified of the publication date in due course.

With kind regards,

Timea Kemeri-Szekernyes

PLOS Genetics

On behalf of:
